# Cohesin Mutations Induce Chromatin Conformation Perturbation of the *H19*/*IGF2* Imprinted Region and Gene Expression Dysregulation in Cornelia de Lange Syndrome Cell Lines

**DOI:** 10.3390/biom11111622

**Published:** 2021-11-02

**Authors:** Silvana Pileggi, Marta La Vecchia, Elisa Adele Colombo, Laura Fontana, Patrizia Colapietro, Davide Rovina, Annamaria Morotti, Silvia Tabano, Giovanni Porta, Myriam Alcalay, Cristina Gervasini, Monica Miozzo, Silvia Maria Sirchia

**Affiliations:** 1Medical Genetics, Department of Health Sciences, Università degli Studi di Milano, 20142 Milano, Italy; silvana.pileggi@unimi.it (S.P.); martalav04@gmail.com (M.L.V.); elisaadele.colombo@unimi.it (E.A.C.); laura.fontana@unimi.it (L.F.); davide.rovina@gmail.com (D.R.); cristina.gervasini@unimi.it (C.G.); silvia.sirchia@unimi.it (S.M.S.); 2Unit of Medical Genetics, ASST Santi Paolo e Carlo, 20142 Milano, Italy; 3Department of Pathophysiology and Transplantation, Medical Genetics, Università degli Studi di Milano, 20122 Milan, Italy; patrizia.colapietro@unimi.it (P.C.); silvia.tabano@unimi.it (S.T.); 4Research Laboratories Coordination Unit, Fondazione IRCCS Ca’ Granda Ospedale Maggiore Policlinico, 20122 Milano, Italy; annamaria.morotti@policlinico.mi.it; 5Laboratory of Medical Genetics, Fondazione IRCCS Ca’ Granda Ospedale Maggiore Policlinico, 20122 Milan, Italy; 6Centro di Medicina Genomica, Department of Medicine and Surgery, Università degli Studi dell’Insubria, 21100 Varese, Italy; giovanni.porta@uninsubria.it; 7Department of Experimental Oncology, IEO European Institute of Oncology IRCCS, 20139 Milan, Italy; myriam.alcalay@unimi.it; 8Department of Oncology and Hemato-Oncology, University of Milan, 20122 Milan, Italy

**Keywords:** cohesin, Cornelia de Lange Syndrome, 3D chromatin conformation, IGF2/H19 domain, WNT pathway, imprinted genes

## Abstract

Traditionally, Cornelia de Lange Syndrome (CdLS) is considered a cohesinopathy caused by constitutive mutations in cohesin complex genes. Cohesin is a major regulator of chromatin architecture, including the formation of chromatin loops at the imprinted *IGF2*/*H19* domain. We used 3C analysis on lymphoblastoid cells from CdLS patients carrying mutations in *NIPBL* and *SMC1A* genes to explore 3D chromatin structure of the *IGF2*/*H19* *locus* and evaluate the influence of cohesin alterations in chromatin architecture. We also assessed quantitative expression of imprinted *loci* and WNT pathway genes, together with DMR methylation status of the imprinted genes. A general impairment of chromatin architecture and the emergence of new interactions were found. Moreover, imprinting alterations also involved the expression and methylation levels of imprinted genes, suggesting an association among cohesin genetic defects, chromatin architecture impairment, and imprinting network alteration. The WNT pathway resulted dysregulated: canonical WNT, cell cycle, and WNT signal negative regulation were the most significantly affected subpathways. Among the deregulated pathway nodes, the key node of the frizzled receptors was repressed. Our study provides new evidence that mutations in genes of the cohesin complex have effects on the chromatin architecture and epigenetic stability of genes commonly regulated by high order chromatin structure.

## 1. Introduction

Cohesin is a chromatin-associated multi-subunit protein complex that is highly conserved during evolution and involved in several aspects of chromosome biology [1], such as cell division, DNA damage repair, gene transcription and chromosome architecture [2,3].

Cohesin is a ring-shaped complex composed of the SMC family proteins, SMC1 (also known as SMC1A) and SMC3, which function by forming heterodimers with two non-SMC components: RAD21 and SCC3. This core subunit orchestrates long-range DNA interactions to mediate sister chromatid cohesion during the cell cycle, essential for accurate chromosome segregation [4]. Other components of the cohesin complex are NIPBL and HDAC8: NIPBL mediates the loading of cohesin on chromatin during S-phase, G1 and G2 [5]. Conversely, the removal of SMC3 from chromatin during prophase and anaphase is mediated by HDAC8, which functions as an SMC3 deacetylase to permit the correct dissolution of pro-cohesive elements and the recycling of “refreshed” cohesin for a new cell cycle [6].

Cohesin is clinically relevant because heterozygous mutations in genes encoding for complex subunits lead to developmental disorders called cohesinopathies, whereas loss of function of the cohesin complex is incompatible with life [7]. Cornelia de Lange Syndrome (CdLS; OMIM #122470, 300590, 610759, 300882, and 614701) is a neurodevelopmental disorder caused by dominant variants in genes encoding structural and regulatory cohesin proteins. CdLS has an estimated occurrence of one in every 10,000–30,000 and is characterized by a peculiar face with arched eyebrows, synophrys, ptosis, upturned nose, thin upper lip and micrognathia, hirsutism, intellectual disability, growth delay and multisystem malformations [2,8,9].

The major causative genes of the Cornelia de Lange Syndrome are *NIPBL, SMC1A, SMC3, RAD21* and *HDAC8*. Mutations in the *NIPBL* gene are responsible for more than 65% of CdLS cases [10,11] and frameshift or nonsense mutations, resulting in *NIPBL* haploinsufficiency, often confer more severe phenotypes compared with missense mutations [12]. Variants in *SMC1* and *SMC3* were found in a minor subset of CdLS cases (~5% and <1%, respectively) showing a milder phenotype, with mental retardation accompanied by other less severe abnormalities [13,14,15]. In addition, mutations in the X-linked gene *HDAC8* are found in a small number of CdLS patients and cause a phenotypically distinct subgroup [14]. In addition to the overmentioned cohesin core complex alterations, mutations in other genes, such as *AFF4*, *ANKRD11*, *CREBBP*, and *EP300*, have been identified in patients with a phenotype resembling CdLS [9].

Overall, since the cohesin complex is involved in regulating gene expression during embryogenesis, cohesinopathies are characterized by a variety of developmental defects, including growth and mental delay, limb deformities, and craniofacial anomalies [16]. In particular, intellectual disability is related to impairment in neuronal development and transcriptional regulation (including initiation, general transcription, elongation, pausing, backtracking, processing, termination, and associated epigenetic modifications) [17].

Cohesin favors long-range DNA interactions and binds to many sites throughout the genome, sometimes in combination with the CCCTC-binding factor (CTCF) insulator protein, which mediates chromatin loop formation [18]. Cohesin and CTCF cooperate in the regulation of gene expression and chromosome structure [3,19,20,21]. Several studies reported that long-range interactions involving regulatory sequences are reduced by cohesin knockdown or cleavage highlighting the involvement of three-dimensional (3D) chromatin organization by cohesin in the regulation of many genes [3,22,23,24].

In particular, at the imprinted *IGF2/H19* domain CTCF plays an important role in organizing allele-specific higher-order chromatin conformation and functions as an enhancer, antagonizing the activity of a transcriptional insulator. The 3D chromatin structure of this domain was extensively studied [21,25]. The CTCF mediated insulator is located upstream of the *H19* gene and is known as the imprinting control region 1 (IC1). (Epi)genetic defects at the *IGF2/H19 locus* are associated with Beckwith–Wiedemann (BWS OMIM #130650) and Silver–Russell (SRS OMIM #180860) syndromes, two imprinted disorders characterized by opposite growth defects [26].

The genomic imprinting process is a parent-of-origin specific mark of the genome, leading to monoallelic expression of a subset of genes. Parental specific monoallelic gene expression of *H19* and *IGF2* imprinted genes (located at 11p15.5 imprinted domain) is controlled by the methylation of IC1 [27], also called the *H19* differentially methylated region (DMR) or domain (DMD). IC1 acquires methylation on the paternal allele during the male gametogenesis. Depletion of either cohesin or CTCF results in reduced transcription of *H19* and increased expression of *IGF2*, implying a role for these proteins in the expression regulation of these *loci* [21,28]. The putative looped structure at the *IGF2/H19* domain, which brings the promoter and enhancer together in a parental allele-specific manner, is dependent on the differential methylation of IC1, and both cohesin and CTCF bind to several regions within the *locus* [29,30]. Recently, we provided a detailed characterization of the chromatin architecture of the 11p15.5 imprinted domain [31]; our data extended the available information regarding the structure of the *IGF2/H19* domain and defined the interactome of the *CDKN1C/KCNQ1OT1* domain (the second imprinted *locus* mapped at 11p15.5) and the long-range contacts involving the two domains. We confirmed that this domain folds in complex chromatin conformations, which facilitate the control of imprinted genes mediated by distant enhancers, and found deep alterations in the chromatin structure of the entire imprinted domain in lymphoblastoid cell lines (LCLs) from BWS and SRS patients.

Nativio and collaborators [29] reported that cohesin depletion by RNAi results in a modest reduction in looping interactions at the imprinted *IGF2/H19* domain, confirming that cohesin is a stabilizing factor in chromatin looping. Although such defects may be very subtle, they hold the potential to cause changes in gene regulation. In light of this evidence, we hypothesized that genetic variations of cohesin genes may also affect the chromatin structure of *loci* that involve cohesin. With this aim, using chromatin conformation capture (3C) we explored whether chromatin structure and methylation of the *IGF2/H19* domain may be impaired in LCLs from CdLS patients carrying different genetic alterations. We also quantified the expression profiles and methylation pattern of a panel of imprinted genes that are often regulated by a high order chromatin structure [32,33]. Expression analysis also included genes belonging to the WNT pathway, reported to be altered in CdLS [34,35,36,37].

## 2. Materials and Methods

### 2.1. Lymphoblastoid Cell Lines

LCLs (Table 1) were generated from nine pediatric CdLS patients carrying mutations in *SMC1A* (CdLS1–5) [38,39] and *NIPBL* (CdLS6–9) [40,41] and four nonaffected aged matched controls (CTRL 1–4). The study was approved by the Ethics Committee of Università degli Studi di Milano (Comitato Etico number 99/20, 17 November 2020). Appropriate written informed consent was obtained from patients’ parents.

### 2.2. Chromatin Conformation Capture Assay (3C)

3C was performed as previously described [31], where the 3C primer details, sequences and the extensive description of the regions analyzed were provided.

Given the importance of CTCF binding at specific sites for the development of intra- and interchromosomal contacts [42,43,44], we analyzed four clusters of CTCF-binding sites in the domain: one upstream of *IGF2* (CTCF Up), one downstream of *H19* (CTCF Down), one in the centrally conserved domain (CCD) region, and one in IC1 (Figure 1a and Rovina et al., 2020 [31]), in addition to genes and enhancers mapped in the 11p15.5 region (Figure 1a).

### 2.3. DNA Methylation Analysis

Total DNA was extracted from LCLs using the QIAamp^®^ DNA Mini kit (Qiagen, Hilden, Germany) according to the manufacturer’s instructions. Two independent quantitative methylation experiments of *IGF2*, *H19*, *GNAS, GNAS-AS1, MEST* and *PEG10* DMRs were performed by pyrosequencing using the Pyro Mark ID instrument (Qiagen, Hilden, Germany). Raw data were analyzed using the Q-CpG software v1.09 (Biotage Sweden AB). Details on the genomic positions, set-up and protocol were previously described [45,46].

### 2.4. nCounter Analysis

Total RNAs were obtained using the Qiazol reagent (Qiagen, Hilden, Germany), followed by RNA purification by the RNeasy mini kit (Qiagen, Hilden, Germany), according to the manufacturer’s protocol. RNAs were eluted in 50 μL of RNase-free water. Concentration and purity were evaluated using the Nanodrop (Thermo Fisher Scientific, Wilmington, DE, USA).

Expression analysis was performed by Ncounter using the Nanostring Vantage 3D^TM^ RNA WNT Pathways Panel (Nanostring, Seattle, WA, USA) using a panel including 180 genes associated with the WNT pathways and 12 reference genes for normalization (*CC2D1B, COG7, EDC3, GPATCH3, HDAC3, MTMR14, NUBP1, PRPF38A, SAP130, SF3A3, TLK2, ZC3H14*), customized with 17 imprinted and imprinted-related genes (Appendix A).

The expression profiles were evaluated starting from 150 ng of total RNA for each sample, which integrity was assessed with the TapeStation 2200 (Agilent, Santa Clara, CA, USA); RNA integrity number (RIN) values > 7.0 were considered suitable for the experiments. We used Nanostring technology as it represents a medium-throughput platform to evaluate mRNA abundance profiles providing reproducible and fully automated analyses of the samples. The robustness of this technology was already validated in several papers [47,48]. The reliability of Nanostring technology is based on the ability to quantify the expression of multiple genes without amplification steps. Conversely, technical artifacts could be introduced in qPCR [49].

### 2.5. Statistical Analysis

3C assays: two independent 3C assays were performed for each sample. The frequencies of associations are expressed as mean ± standard deviation. The control value is derived from four independent 3C assays, two using CTRL1 and two CTRL2. The mean of each CdLS cell line is calculated from the results of two independent 3C assays. Differences in association frequencies between controls and patient’s LCLs were evaluated using the two-way ANOVA test followed by the Bonferroni post-test in the GraphPad Prism program. Statistical significance is indicated as **** *p* ≤ 0.0001; *** *p* ≤ 0.001; ** *p* ≤ 0.01; * *p* ≤ 0.05.

NCounter analysis: Nanostring data were analyzed by the nSolver Advanced Analysis Software 4.0 (NanoString, Seattle, WA, USA) considering a background threshold of 20 counts and excluding from the analysis all genes with counts above the threshold. Quality assessment was performed for each sample, and two quality control parameters common to all nCounter assays were considered: the Imaging QC that measures the percentage of the requested fields of view successfully scanned in each cartridge lane and the Binding Density QC that measures the reporter probe density on the cartridge surface in each sample lane. The Benjamini–Hochberg method was applied to reduce the false discovery rate (FDR), minimizing Type I errors (false positives). Unadjusted *p*-value ≤ 0.05 were considered significant.

## 3. Results

A schematic overview of the experimental design is summarized in Table 1.

### 3.1. Chromatin Interactions at the IGF2/H19 Domain in Cells from CdLS Patients

Nativio and coworkers [29] showed a reduction in the looping interactions at the *IGF2*/*H19* imprinted domain after cohesin depletion by RNAi, suggesting a causative relationship between cohesin anomalies and deregulation of imprinted genes. We investigated whether genetic alterations (point mutations and balanced reciprocal translocations) in *SMC1A* and *NIPBL* affect the chromatin structure of the *IGF2*/*H19 locus*. The detailed landscape of the *IGF2*/*H19* region analyzed by 3C and the 3C coverage is depicted in Figure 1a.

To study the physical contacts in the region, we used four anchors: CTCF Up, IC1, Enh A and CTCFDw (Figure 1). Using this approach, we recently characterized the 3D chromatin structure of the region in BWS and SRS patients and in normal cells [31]. The interactome of the control LCLs included in this study is schematically shown in Figure 1b.

We performed 3C analysis in four CdLS LCLs with genetic defects in *SMC1A* (CdLS1 and 2) and *NIPBL* (CdLS6 and 9) genes (Table 1) to identify modifications of the chromatin interactome in the IC1 domain. Two independent experiments were carried out for each cell line. The specific contacts among CTCF-binding sites, regulatory elements and genes mapped in the domain responsible for the repositioning of the regional enhancers near *IGF2* or *H19* in normal conditions, are summarized in the interactome scheme shown in Figure 2a, Figure 3a and Figure 4a, and the details for each anchor are provided in Figure 2b, Figure 3b, Figure 4b and Appendix A.

We found an aberrant chromatin structure of the *IGF2/H19* region in CdLS cells compared to controls (Figure 2, Figure 3, Figure 4 and Appendix A). In particular, interactions of the *IGF2* promoter and the CTCF Up were partially conserved, while those involving the enhancer A and the CTCF Dw were perturbed, especially in the CdLS1 and CdLS6 LCLs (Figure 2 and Figure 3). In addition, new interactions between the CCD and the *H19* region (IC1, Enh A and CTCF Dw), and between 3′ *IGF2* and the IC1/CTCF Up, were observed in all the CdLS LCLs (Figure 2, Figure 3, Figure 4 and Appendix A).

In summary, 3C results show a general impairment of the chromatin architecture in CdLS cells regardless of the specific cohesin gene involved. The emergence of many new interactions could be a further expression of the malfunction of the cohesin complex.

### 3.2. Expression Profile of the CdLS Cell Lines

Using a Nanostring approach, we analyzed the expression levels of a set of imprinted genes, related-imprinted genes (Table 2) and *loci* of the WNT pathway (Table 3) in nine CdLS LCLs with different mutations in *SMC1A* (CdLS 1–5) or *NIPBL* (CdLS 6–9) genes, and in four control LCLs (CTRL 1–4) (Table 1).

### 3.3. Imprinted and Imprinted-Related Genes Panel

Given that genetic defects in cohesin genes were found to be associated with a general impairment of the 3D chromatin structure of the *IGF2*/*H19* imprinted domain, we investigated the expression profiles in CdLS LCLs in a panel of imprinted genes that are often regulated by a high order of chromatin structure [32,33]. We evaluated the expression levels of 13 imprinted genes and four imprinted-related genes (Appendix A). The analysis of the imprinted genes showed that 7 out 13 were expressed in LCLs (Table 2), and no expression in LCLs was retrieved for *IGF2* and *H19*.

Three out of seven imprinted genes were differentially expressed (DEGs) in patients’ cell lines. In particular, *GNAS* was upregulated, whereas *MEST* and *PEG10* were downregulated. Furthermore, *GNAS-AS1* was downregulated despite being close to the threshold of significance (unadjusted *p*-value = 0.0058) (Table 2 and Figure 5).

These results indicate that the imprinting network was altered, suggesting an association among cohesin defects, 3D chromatin architecture impairment and imprinting network alteration.

### 3.4. WNT Panel

Diverse WNT pathways act as master regulators of central nervous system development [35], which is disrupted in CdLS animal models and patient-derived cells lines [35,36,37,50]. Therefore, we next analyzed a panel of 180 genes involved in WNT signaling (Appendix A), of which 163 were expressed in LCLs.
Figure 5Volcano plot of DEGs CdLS compared to control cell lines. Upregulated genes are highlighted by red dots, while downregulated genes by blue dots. In bold and underlined are reported the DE imprinted genes. FDR < 0.1 (Benjamini–Hochberg adjusted *p* value) and unadjusted *p* value < 0.05 are indicated by horizontal lines. The VolcaNoser tool was used for creating volcano plots [51].
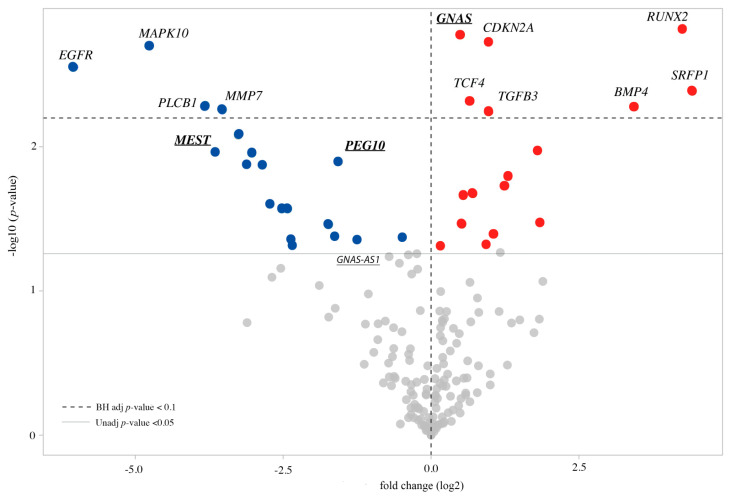


Thirty-three differentially expressed (DE) genes were observed in patients’ LCLs (unadjusted *p*-value < 0.05) compared to controls. Among them, 16 were upregulated and 17 downregulated (Figure 5 and Table 3). Pathway enrichment analysis by the nSolver software indicated that the most significant differences observed in CdLSs cells are in the following subpathways: canonical WNT pathway, cell cycle and WNT signal negative regulation (Figure 6). Among the investigated genes, an altered expression was mainly observed in members of the WNT receptors of the *Frizzled* gene (*FZD*) family *(FZD5, FZD8* and *FZD10* downregulated and *FDZ3* upregulated).

Our results confirm alterations of the WNT pathways in LCLs derived from CdLS patients; in particular, we observed the most significant differences in the canonical WNT pathway, cell cycle and WNT signal negative regulation. Figure 7 shows a schematic representation of the expression alterations observed in the three main WNT signaling pathways. The nodes of the pathways that were found to be dysregulated by Pathview (nSolver Advanced Analysis Software 4.0; Figure 7) were: Frizzled (DEGs: *FZD3*, *FZD5*, *FZD8* and *FZD10*), Wnt (DEG: *WNT10B*), FRP (DEG: *SFRP1*), GBP (DEG: *FRAT1*), TCF/LEF (DEGs: *TCF4*, *LEF1* and *TCF7L1*), Gro/TLE (DEG: *TLE1*), Uterine (DEG: *MMP7*), JNK (DEG: *MAPK10*), PLC (DEG: *PLCB1*), and CaN (DEGs: *PPP3CA* and *PPP3CC*).

Finally, we performed three different expression analyses splitting CdLSs samples based on *NIPBL* and *SMC1A* involvement, comparing *NIPBL*-mutated CdLSs vs. CTRLs (Appendix A), *SMC1A*-mutated CdLSs vs. CTRLs (Appendix A) and *NIPBL*- vs. *SMC1A*- mutated CdLSs (Appendix A). Differently to the main cohort (CdLSs vs. CTRLs, Figure 5 and Figure 6), in none of the three subsets, we found differentially expressed genes with Benjamini–Hochberg adjusted *p*-values <0.1. Some genes reached a significant unadjusted *p*-value < 0.005. Interestingly, considering the *NIPBL*-mutated samples, 17 DEGs reached the significant threshold of 0.005, among them nine are shared with the most significant DEGs of the main cohort: *SFRP1, TCF4, GNAS, MEST, MAPK10, CDKN2A, PLCB1, RUNX2* and *MMP7* (Appendix A). Similarly, the *SMC1A*-mutated group showed nine out of the 25 significant DEGs in common with the main cohort: *RUNX2, BMP4, MAPK10, GNAS, CDKN2A, TGFB3, TCF4, PEG10* and *PLGB1* (Appendix A). When we consider the *NIPBL*- vs. *SMC1A*-mutated CdLSs subset, the number of shared significant DEGs (*p*-value < 0.005) drops to 2: *RUNX2* and *MAPK10* (Appendix A). These findings are in line with the results of Boudaoud et al. that compared the gene expression profiles of LCLs from patients carrying mutations in *NIPBL* and *SMC1A* and showed reduced number of differentially expressed genes shared between the two gene groups (*n* = 126) compared to the totality of genes misregulated in *NIPBL*-mutated (*n* = 1431) and *SMC1A*-mutated (*n* = 1186) samples [52].

### 3.5. Methylation Analysis of the Imprinted DMRs

We analyzed the methylation status of the *IGF2*/*H19*, *MEST, PEG10, GNAS* and *GNAS-AS1* DMRs by pyrosequencing to evaluate possible correlations between the observed chromatin structure and expression defects and the methylation levels of the regulatory regions of these imprinted genes.

In CdLS LCLs we observed a general instability in the methylation status of the IC1 and *MEST*-DMR, a trend to hypermethylation of the *PEG10* and *GNAS-AS1* DMRs and a trend to hypomethylation of the *GNAS*-DMR compared to CTRLs (Figure 8 and Appendix A): IC1 methylation range was 9–45% in CdLSs and 34–41% in controls; *MEST*-DMR methylation range was 33–60% in CdLSs and 42–47% in controls; *PEG10*-DMR methylation range was 41–44% in CdLSs and 35–43% in controls; *GNAS*-DMR methylation range was 4–42% in CdLSs and 31–37% in controls; *GNAS-AS1*-DMR methylation range was 8–37% in CdLSs and 10–22% in controls. In CdLS cells, hypermethylation of *PEG10*, *MEST* and *GNAS-AS1* DMRs was in line with the observed low expression, in the same way *GNAS*-DMR hypomethylation was in line with its overexpression; the alterations in IC1 methylation may be the result of the perturbation in chromatin architecture of the *IGF2*/*H19* imprinted domain. Regarding the specific gene mutated in the CdLS LCLs a more pronounced variation in methylation status is apparent in *SMC1A* involvement, in particular for *MEST, PEG10* and *GNAS-AS1* DMRs.

## 4. Discussion

Cohesin has complex functions in chromosome biology, including gene expression regulation and maintenance of chromatin architecture [53]. Defects in cohesin function are associated with a group of diseases known as cohesinopathies, notably CdLS.

Starting from the evidence that cohesin depletion causes a reduction in the looping interactions in the *IGF2*/*H19* imprinted domain [29], we studied the 3D chromatin structure of this *locus* in LCLs from CdLS patients with mutations in the *SMC1A* or *NIPBL* genes, and evaluated whether constitutive genetic mutations in the cohesin subunit genes can alter chromatin architecture and, consequently, gene expression.

We found a broad perturbation in the chromatin structure of the domain regardless of the CdLS causative gene, and observed a change in the interactions among CTCF-binding sites, regulatory elements and genes of the region. This scenario could be related to the existence in each cell line of a diffuse instability rather than recurrent alterations of specific interactions. It is also conceivable that compromised maintenance of chromatin architecture could lead to heterogeneous defects among cells in the same cell line. The alterations can be due to cohesin malfunction rather than lack of function. The improper sliding of the cohesin complex along the loop could, indeed, cause the observed perturbed chromatin interactions, including new associations.

In addition, we found that the 3D chromatin alterations of the *locus* were associated with methylation defects of IC1 in some CdLS cell lines. Methylation of this DMR controls the expression of the *IGF2* and *H19* imprinted genes. In our recent study we reported that, in BWS and SRS imprinting-related disorders, the causative methylation defects at IC1 are associated with alterations in the 3D chromatin architecture of the domain [31]. Differently, in CdLS it is conceivable that the observed methylation instability is related to chromatin structure alterations caused by a malfunction of the cohesin complex.

Imprinting disorders are frequently associated with growth and development abnormalities [54,55]; similarly, prenatal and postnatal growth restriction are observed in CdLS patients [56]. This feature might be also related to an imprinting network perturbation suggested by our data. In particular, we found methylation instability of the DMRs and/or defective expression of *loci* involved in fetal growth such as *PEG10*, *MEST*, *GNAS*, *IGF2* and *H19* [55,56,57].

DMRs methylation instability at multiple imprinted *loci* has already been identified in patients with multilocus imprinting disturbances (MLID), thus demonstrating a fine-tuned network involving regulatory regions of imprinting [58]. Similarly, in CdLS we surmise a broad impairment of the maintenance of the regulatory mechanisms of genomic imprinting, resulting in a general instability of the imprinting marks, which might be related, in this disease, to cohesin defects. These findings strengthen the key role of this complex in transcription regulation.

Based on the pivotal role of cohesin in the dynamics of the transcriptional regulation network, and on the hypothesis that the multiorgan defects in CdLS patients are due to global disruption of the transcriptional regulation network of developmental pathways, including the canonical WNT pathway [59,60,61], we investigated the expression profile of a panel of 180 genes of the WNT pathways. In vertebrates, the WNT signaling pathway regulates crucial aspects of the embryonic development and maintenance of adult tissue homeostasis by regulating cell proliferation, differentiation, migration, genetic stability, and apoptosis, as well as maintenance of adult stem cells in a pluripotent state [62]. These processes are obtained through two principal branches: the canonical pathway that regulates the expression of the key developmental target genes through the frizzled family receptors and, the intracellular transducer Dishevelled, and the noncanonical WNT pathway (β-catenin-independent pathway) that regulates cell polarity and dorsal mesodermal cell movements during development [37,63].

We found 33 genes specifically deregulated in CdLS LCLs, some of which are involved in transcriptional regulation: *RUNX2*, *CDKN2A, TCF4, LEF1, SOX2* and *TCF7L1*.

*RUNX2* gene was the most significantly upregulated; it is a member of the *RUNX* family of transcription factors, nuclear proteins with a DNA-binding domain. *RUNX2* is essential for osteoblastic differentiation and skeletal morphogenesis and acts as a scaffold for other regulatory factors involved in tissue-specific expression of the skeletal. Mutations in this gene have been associated with cleidocranial dysplasia [64,65]. The most strongly downregulated gene was *MAPK10,* a gene involved in neuronal proliferation, differentiation and survival. Interestingly, *RUNX2* and *MAPK10* are the only two significant DEGs (unadjusted *p*-value 0.0462 and 0.0221, respectively) when we consider the *NIPBL*- vs. *SMC1A*- mutated CdLSs subset.

Our results confirm alterations of the WNT pathways in LCLs derived from CdLS patients; in particular, we observed the most significant differences in the canonical WNT pathway, cell cycle and WNT signal negative regulation. In addition, we found that several genes belonging to the pathway nodes were deregulated; in particular the key node of the frizzled receptors was repressed, specifically *FDZ3, FZD5, FZD8* and *FZD10* genes.

Because cohesin represents the primary regulator of 3D genome organization and, consequently, of the gene expression in all cell types, we used LCLs as a model for CdLS; however, we are aware that different expression profiles may be present in cells of other embryological origin.

Our study provides new evidence that the mutations in genes of the cohesin complex may affect chromatin architecture and the epigenetic stability of genes commonly regulated by high order chromatin structures, such as the imprinted *loci*. Altered imprinted gene expression could, at least in part, explain the growth defects present in CdLS.

## Figures and Tables

**Figure 1 biomolecules-11-01622-f001:**
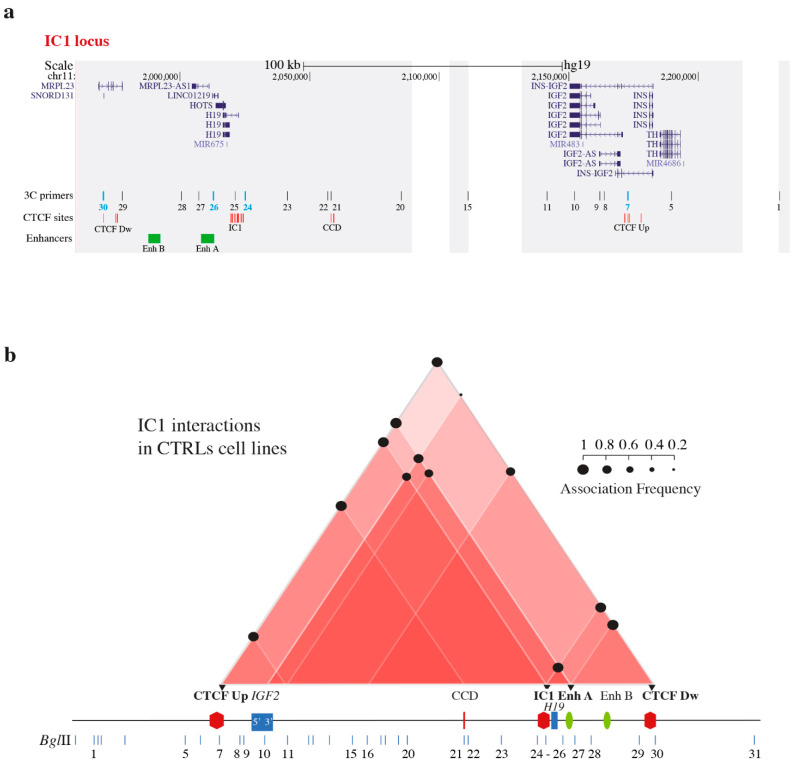
3C and interactome analyses in control cell lines. (**a**) Scheme of the IC1 *locus* under analysis (UCSC Genome Browser map position 1,960,000 to 2,235,000). Areas covered by 3C analysis are highlighted in grey. The locations of genes are indicated in the upper part. Vertical black lines, corresponding to BglII restriction sites, indicate the primers used for 3C analysis. Anchor primers are highlighted in turquoise. Red and green lines indicate clusters of CTCF-binding sites in reverse or forward orientation, respectively. Green bars (EnhA and EnhB) correspond to enhancer regions. CTCF-binding sites: CTCF Up, CCD, IC1, and CTCF Dw. (**b**) Schematic representation of the IC1 domain interactome in control cell lines. The data for each cell line were derived from two independent 3C experiments. Interactions between different elements of the IC1 region are shown by red triangles; increasing color intensity corresponds to an increase in the number of interactions of the subregion. Mean association frequencies of CTRL1 and CTRL2 are indicated with black circles. Black triangles indicate the anchors used for 3C analysis. A linear representation of the domain is shown below. In accordance with the transcription of genes in the region, panel b was drawn in reverse orientation with respect to the map in panel a. Adapted from [31] following the Creative Commons Attribution 4.0 International License.

**Figure 2 biomolecules-11-01622-f002:**
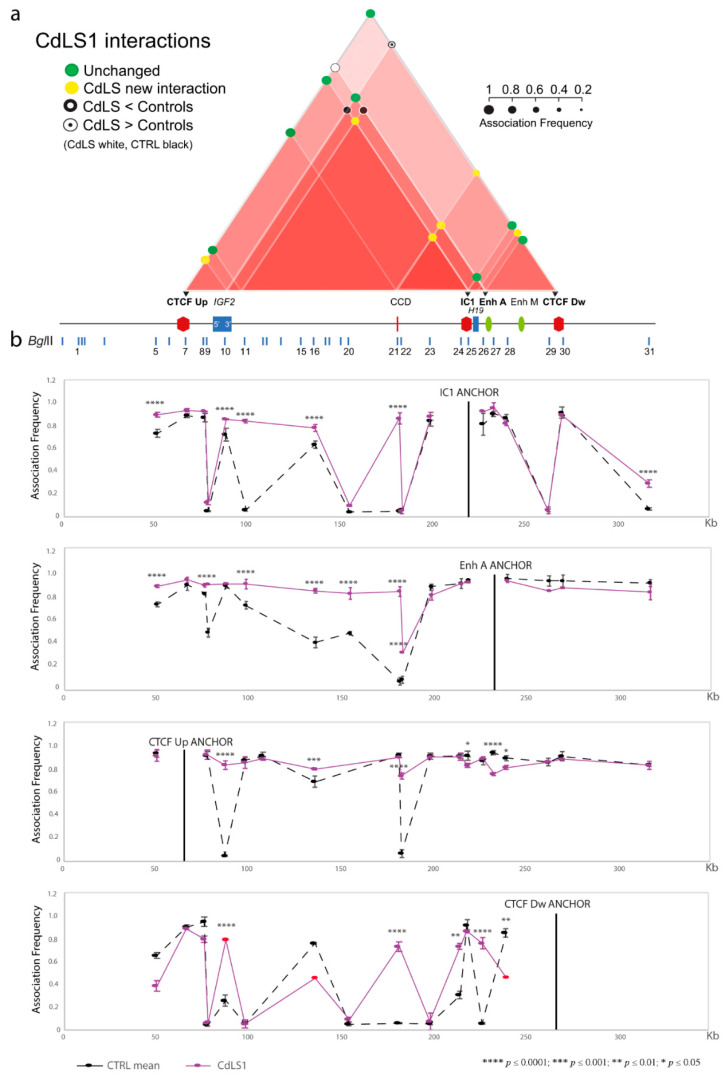
Abnormalities in chromatin architecture at the IC1 *locus* of the CdLS1 cell line (*SMC1A* mutation). The figure is to scale. (**a**) Scheme of the statistically significant modifications in the chromatin interactome of the IC1 domain in the CdLS1 cell line compared with the mean of the controls. Interactions between different elements of the IC1 region are shown by red triangles; increasing color intensity corresponds to an increase in the number of interactions of the sub-region. Colored circles represent association frequencies as described in the figure. Black triangles indicate the anchors used for 3C analysis. A linear representation of the IC1 imprinted domain is depicted below the interactome. (**b**) IC1 *locus* looping profiles for the indicated anchors in controls (dotted black) and CdLS1 (pink) cell lines. BglII restriction sites are indicated above. Each point in the profile is the mean ± standard deviation of two independent 3C experiments and indicates the association frequency between the anchor and the fragment on the left of the corresponding BglII restriction site. Differences (two-way ANOVA test) are indicated by asterisks. Red dots indicate the points with standard deviation > 0.1.

**Figure 3 biomolecules-11-01622-f003:**
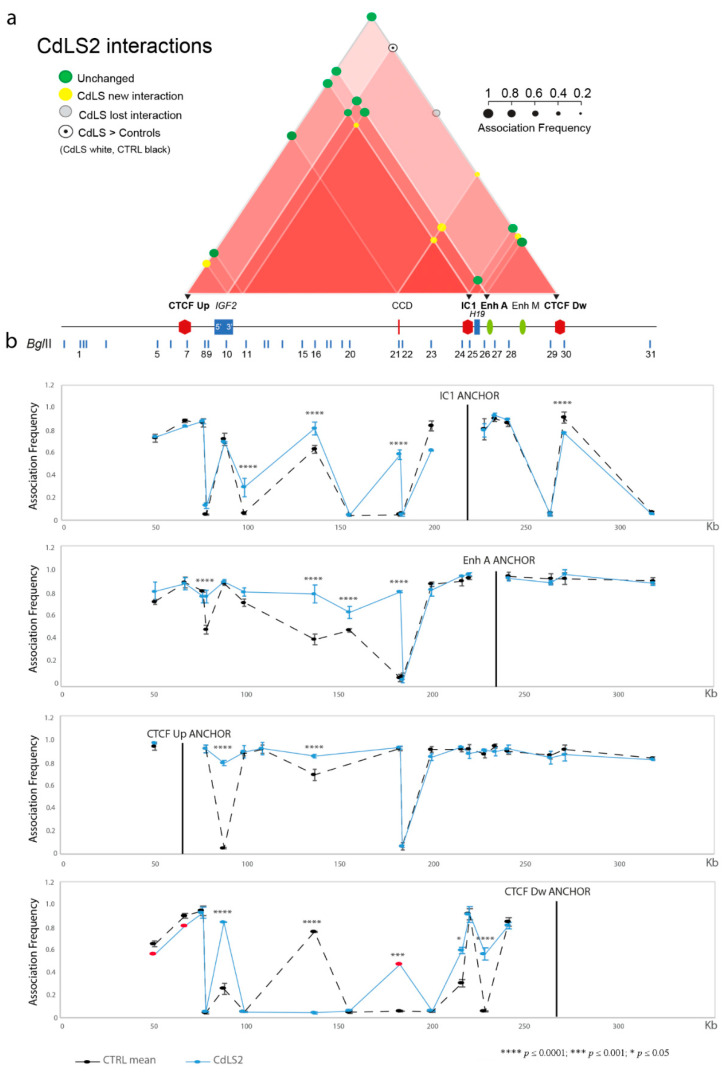
Abnormalities in chromatin architecture at the IC1 *locus* of the CdLS2 cell line (*SMC1A* mutation). The figure is to scale. (**a**) Scheme of the statistically significant modifications in the chromatin interactome of the IC1 domain in the CdLS2 cell line compared with the mean of the controls. Interactions between different elements of the IC1 region are shown by red triangles; increasing color intensity corresponds to an increase in the number of interactions of the subregion. An interaction lost selectively in the CdLS2 cell line is shown in light grey. Colored circles represent association frequencies as described in the figure. Black triangles indicate the anchors used for 3C analysis. A linear representation of the IC1 imprinted domain is depicted below the interactome. (**b**) IC1 *locus* looping profiles for the indicated anchors in controls (dotted black) and CdLS2 (light blue) cell lines. BglII restriction sites are indicated above. Each point in the profile is the mean ± standard deviation of two independent 3C experiments and indicates the association frequency between the anchor and the fragment on the left of the corresponding BglII restriction site. Differences (two-way ANOVA test) are indicated by asterisks. Red dots indicate the points with standard deviation > 0.1.

**Figure 4 biomolecules-11-01622-f004:**
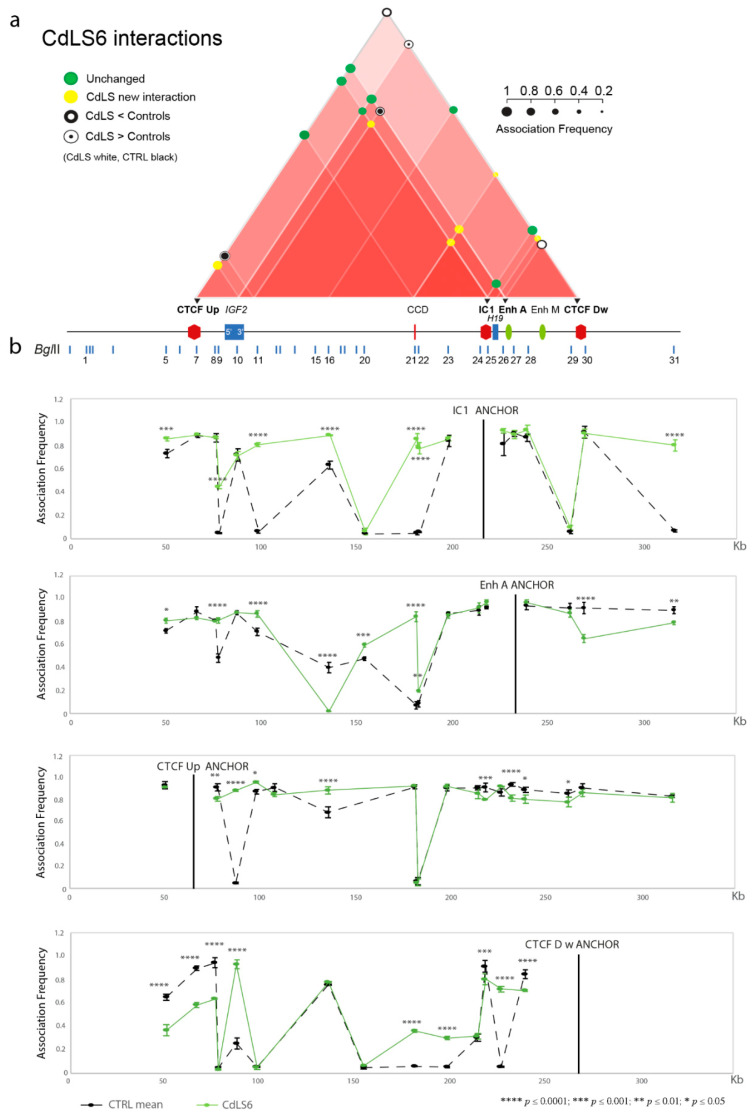
Abnormalities in chromatin architecture at the IC1 *locus* of the CdLS6 cell line (*NIPBL* mutation). The figure is to scale. (**a**) Scheme of the statistically significant modifications in the chromatin interactome of the IC1 domain in the CdLS6 cell line compared with the mean of the controls. Interactions between different elements of the IC1 region are shown by red triangles; increasing color intensity corresponds to an increase in the number of interactions of the subregion. Colored circles represent association frequencies as described in the figure. Black triangles indicate the anchors used for 3C analysis. A linear representation of the IC1 imprinted domain is depicted below the interactome. (**b**) IC1 *locus* looping profiles for the indicated anchors in controls (dotted black) and CdLS6 (green) cell lines. BglII restriction sites are indicated above. Each point in the profile is the mean ± standard deviation of two independent 3C experiments and indicates the association frequency between the anchor and the fragment on the left of the corresponding BglII restriction site. Differences (two-way ANOVA test) are indicated by asterisks.

**Figure 6 biomolecules-11-01622-f006:**
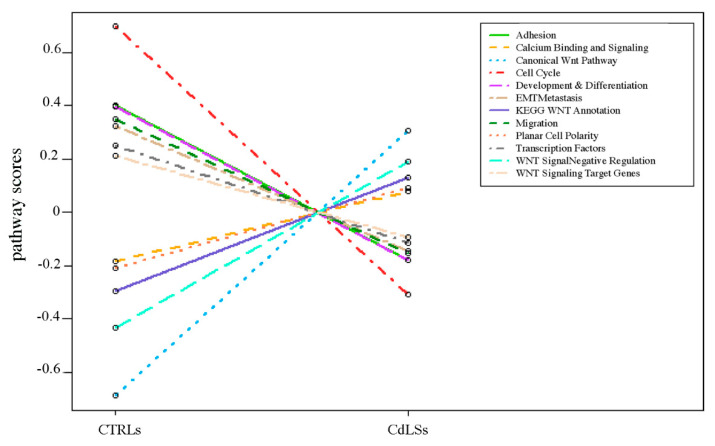
Trend plot of pathway scores vs. sample types (CTRLs and CdLSs). Pathway enrichment analysis was performed by nSolver software (figure rendered by Pathview, nSolver Advanced Analysis Software 4.0).

**Figure 7 biomolecules-11-01622-f007:**
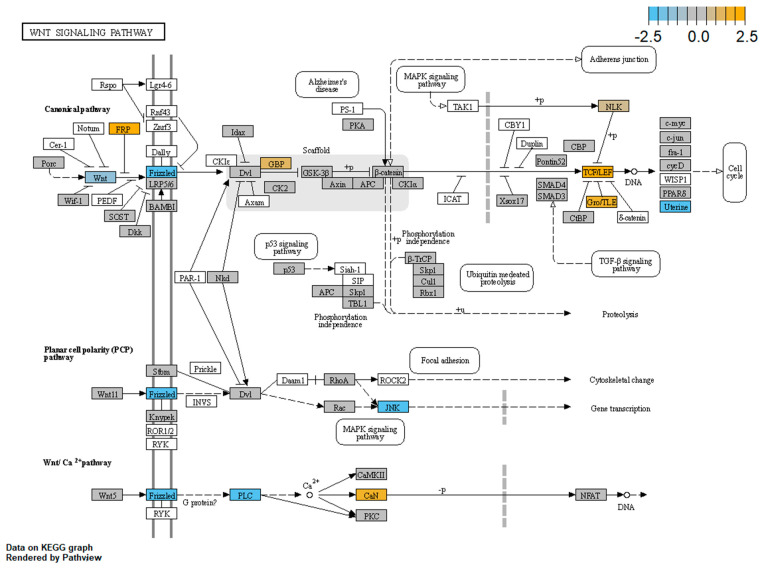
Schematic representation of DEGs in the CdLS cell lines belonging to the three main WNT pathways (figure rendered by Pathview, nSolver Advanced Analysis Software 4.0). Pathway nodes shown in white have no genes in the Vantage 3DTM RNA WNT Pathways Panel. Pathway nodes in grey have corresponding genes in the panel, however no significant differential expression is observed. Nodes in blue and orange denote downregulation or upregulation in CdLSs compared to CTRLs. Node Frizzled, DEGs: *FZD3*, *FZD5*, *FZD8* and *FZD10*; node Wnt, DEG: *WNT10B*; node FRP, DEG: *SFRP1*; node GBP, DEG: *FRAT1*; node TCF/LEF, DEGs: *TCF4*, *LEF1* and *TCF7L1*; node Gro/TLE, DEG: *TLE1*; node Uterine, DEG: *MMP7*; node JNK, DEG: *MAPK10*; node PLC, DEG: *PLCB1*; node CaN, DEGs: *PPP3CA* and *PPP3CC*.

**Figure 8 biomolecules-11-01622-f008:**
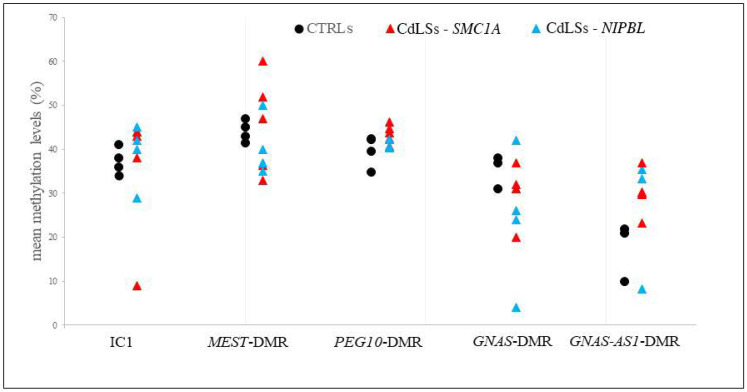
Quantitative CpGs methylation analysis of *IGF2*/*H19* (IC1), *MEST, PEG10, GNAS* and *GNAS-AS1* DMRs from CTRL (black circles) and CdLS cell lines (red triangles: *SMC1A* mutation; light blue triangles: *NIPBL* mutation). Results are the mean of two independent pyrosequencing experiments.

**Table 1 biomolecules-11-01622-t001:** Schematic overview of the experimental design.

Cell Line	Causative Genetic Alteration	Dmr Methylation Analysis			3C Assay	Ncounter Analysis
		IC1	*MEST*-DMR	*PEG10*-DMR		
CTRL1		+	+	+	+	+
CTRL2		+	+	+	+	+
CTRL3		+	+	+	−	+
CTRL4		+	+	+	−	+
CdLS1	*SMC1A* c.2351T > C	+	+	+	+	+
CdLS2	*SMC1A* c.173del16	+	+	+	+	+
CdLS3	*SMC1A* c.2351T > C	+	+	+	−	+
CdLS4	*SMC1A* c.3497A > C	+	+	+	−	+
CdLS5	*SMC1A* c.2078G > A	+	+	+	−	+
CdLS6	*NIPBL* c.-75_ + 65del	+	+	+	+	+
CdLS7	*NIPBL* c.4253G > A	+	+	+	−	+
CdLS8	*NIPBL* c.231-1_231-2del	+	+	+	−	+
CdLS9	*NIPBL* t(5;15)	+	+	+	+	+

Both controls and patient-derived LCLs were cultured in RPMI 1640 medium supplemented with 10% fetal bovine serum (EuroClone, Milano, Italy) and antibiotics (antibiotic-antimycotic 100×, EuroClone, Milano, Italy) at 37 °C in 5% CO_2_.

**Table 2 biomolecules-11-01622-t002:** nCounter Nanostring expression profile of the imprinted genes expressed in LCLs.

Gene	Accession	Log2 Fold Change	*p*-Value	BH *p*-Value
*GNAS **	NM_080425.1	0.494	0.00166	0.0887
*MEST **	NM_177525.1	−3.65	0.0109	0.13
*EG10 **	NM_001040152.1	−1.56	0.0127	0.131
*GNAS-AS1*	NR_002785.2:1026	−0.709	0.0578	0.256
*KCNQ1OT1*	NR_002728.2:31875	−0.535	0.0643	0.277
*FAM50B*	NM_012135.1:1272	0.197	0.289	0.594
*PLAGL1*	NM_006718.3:1872	0.222	0.414	0.698

* The asterisks mark the three differentially expressed imprinted genes. Unadjusted *p*-values are reported in ‘*p*-value’ column and values ≤ 0.05 were considered significant. The Benjamini–Hochberg (BH) method was applied to reduce the false discovery rate (FDR), and the adjusted values are reported in the ‘BH *p*-value’ column.

**Table 3 biomolecules-11-01622-t003:** Expression profile of the genes of the WNT pathway evaluated by nCounter Nanostring technology.

Gene	Accession	Log2 Fold Change	*p*-Value	BH *p*-Value	Pathway Annotation
*RUNX2*	NM_004348.3	4.25	0.00151	0.0887	Development & Differentiation, Transcription Factors
*CDKN2A*	NM_000077.3	0.971	0.00187	0.0887	Cell Cycle, Development & Differentiation, Transcription Factors
*MAPK10*	NM_002753.2	−4.76	0.002	0.0887	KEGG WNT Annotation
*EGFR*	NM_201282.1	−6.04	0.0028	0.0919	Adhesion, Calcium Binding and Signaling, Cell Cycle, Development & Differentiation, Migration
*SFRP1*	NM_003012.3	4.41	0.00411	0.0919	Canonical Wnt Pathway, KEGG WNT Annotation, WNT Signaling Negative Regulation
*TCF4*	NM_003199.1	0.661	0.00481	0.0919	Transcription Factors
*BMP4*	NM_001202.3	3.43	0.00524	0.0919	Development & Differentiation
*PLCB1*	NM_182734.1	−3.82	0.00524	0.0919	KEGG WNT Annotation
*MMP7*	NM_002423.3	−3.53	0.00556	0.0919	Calcium Binding and Signaling, KEGG WNT Annotation, Proteolysis, WNT Signaling Target Genes
*TGFB3*	NM_003239.2	0.978	0.00571	0.0919	Development & Differentiation
*PTGS2*	NM_000963.1	−3.26	0.00834	0.123	Calcium Binding and Signaling, Cell Cycle
*TLE1*	NM_005077.3	1.8	0.0106	0.13	WNT Signaling Negative Regulation
*FZD5*	NM_003468.2	−3.03	0.011	0.13	Canonical Wnt Pathway, KEGG WNT Annotation
*CXCL12*	NM_000609.5	−3.11	0.0132	0.131	EMTMetastasis
*GDNF*	NM_000514.2	−2.85	0.0133	0.131	Development & Differentiation, Migration
*IRS1*	NM_005544.2	1.3	0.016	0.149	Migration
*FRAT1*	NM_005479.3	1.25	0.0185	0.164	Canonical Wnt Pathway, KEGG WNT Annotation
*FZD3*	NM_017412.2	0.699	0.0211	0.173	Canonical Wnt Pathway, KEGG WNT Annotation
*PPP3CC*	NM_005605.4	0.552	0.0216	0.173	KEGG WNT Annotation
*SNAI2*	NM_003068.3	−2.73	0.0249	0.191	EMTMetastasis
*KREMEN1*	NM_001039570.1	−2.52	0.0268	0.191	WNT Signaling Negative Regulation
*FZD10*	NM_007197.2	−2.43	0.027	0.191	KEGG WNT Annotation
*LEF1*	NM_016269.3	1.83	0.0334	0.22	Canonical Wnt Pathway, KEGG WNT Annotation, Transcription Factors
*NLK*	NM_016231.2	0.519	0.0342	0.22	KEGG WNT Annotation, WNT Signaling Negative Regulation
*SOX2*	NM_003106.2	−1.74	0.0348	0.22	Cell Cycle, Development & Differentiation, Transcription Factors
*PPP3CA*	NM_000944.4	1.06	0.0401	0.237	KEGG WNT Annotation
*CXCR4*	NM_003467.2	−1.63	0.0418	0.237	EMTMetastasis
*BIRC5*	NM_001168.2	−0.491	0.0427	0.237	Cell Cycle
*WNT10B*	NM_003394.2	−1.26	0.0442	0.237	KEGG WNT Annotation
*SERPINE1*	NM_001165413.1	−2.36	0.0442	0.237	EMTMetastasis
*TCF7L1*	NM_031283.1	0.931	0.0477	0.24	Canonical Wnt Pathway, KEGG WNT Annotation, Transcription Factors
*FZD8*	NM_031866.1	−2.35	0.0479	0.24	Canonical Wnt Pathway, KEGG WNT Annotation
*SMAD2*	NM_005901.5	0.165	0.0488	0.24	EMTMetastasis, KEGG WNT Annotation

## Data Availability

Not applicable.

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
