# Peer review of "Cohesin Mutations Induce Chromatin Conformation Perturbation of the H19/IGF2 Imprinted Region and Gene Expression Dysregulation in Cornelia de Lange Syndrome Cell Lines"

_biomolecules, 2021, doi:10.3390/biom11111622_

Round 1

Reviewer 1 Report

To the authors,

The article “Cohesin mutations induce chromatin conformation perturbation of the H19/IGF2 imprinted region and gene expression dysregulation in Cornelia de Lange syndrome cell lines” is a clear and well-structured manuscript. However, I have some questions about it.

Question 1: Why do you not carry out the 3C experiment in all the cell lines? Although there is a general impairment of the chromatin structure in the CdLS samples, there are some subtle differences between genes; could you explain that, if there is a reason, apart from the different carried mutation?

Question 2: In the result section “Expression profile of the CdLS cell line” (page 13 line 268), you say that you have analyzed the expression level of a set of imprinted genes and you direct the reader to table 1. However in the table 1 there are not any gene and you do not provided any result of the experiments carried out to fill this section. Could you please provide the data or re-direct to the correct table if there is one?

Question 3: Please, re-write the legend of the table 2 in order to improve its understanding. Although it is written in the main text, it is a bit confusing.

Question 4: Why do you not validate the different expression level results of the WNT panel by qPCR? Please do it to positively confirm the results, at least for the most differently expressed genes.

Question 5: Have you analyzed the data from the WNT panel splitting by gene? NIPBL mutated vs SMC1A mutated samples? In the already published manuscript of Boudaoud I et al from 2017 (Genetics, 2017 Sep;207(1):139-151. doi: 10.1534/genetics.117.202291) they reported that there are only 126 differently expressed genes shared between NIPBL and SMC1A in mutated LCLs derived samples. Please re-analyze our data in this way in order to confirm that these genes are truly dysregulated in both types of samples.
Question 6: I am aware of the difficulty to obtain biological samples of a different origin in CdLS patients; however it will improve the quality of your data if you could confirm the results of some of these experiments in those different samples. 

Best regards,

Author Response

1) Why do you not carry out the 3C experiment in all the cell lines? Although there is a general impairment of the chromatin structure in the CdLS samples, there are some subtle differences between genes; could you explain that, if there is a reason, apart from the different carried mutation?

 Author response: Thank you for this comment that allows us to better explain an important concept that emerged in our study which is the diffuse instability in chromatin architecture. The results obtained in the four cell lines revealed a general impairment of the chromatin structure that is not specific, even considering the specific gene involved in the cell line. This picture could be related to the existence in each cell line of a diffuse instability rather than recurrent alterations of specific interactions. It is also conceivable that compromised maintenance of chromatin architecture could lead to heterogeneous defects among cells in the same cell line. This scenario can be due to cohesin malfunction rather than lack of function. We understand the question of the referee; however, based on the high variability observed among the four analyzed cell lines, we considered that our findings could be sufficient without extending the same 3C experiment to additional cell lines. Our data could be the starting point for further studies aimed at verifying at the single cell level the random disruption of cohesion at different chromatin sites.

We stressed this concept modifying the sentence in the discussion (page 25-26 lines 446-454) as follow:

Before modification, “We found a broad perturbation in the chromatin structure of the domain regardless of the CdLS causative gene and observed a change in the interactions among CTCF-binding sites, regulatory elements and genes of the region. These alterations can be an indicator of cohesin malfunction. The improper sliding of the cohesin complex along the loop could, indeed, cause the observed perturbed chromatin interactions.”

After: “We found a broad perturbation in the chromatin structure of the domain regardless of the CdLS causative gene, and observed a change in the interactions among CTCF-binding sites, regulatory elements and genes of the region. This scenario could be related to the existence in each cell line of a diffuse instability rather than recurrent alterations of specific interactions. It is also conceivable that compromised maintenance of chromatin architecture could lead to heterogeneous defects among cells in the same cell line. The alterations can be due to cohesin malfunction rather than lack of function. The improper sliding of the cohesin complex along the loop could, indeed, causes the observed perturbed chromatin interactions, including new associations.”

2) In the result section “Expression profile of the CdLS cell line” (page 13 line 268), you say that you have analyzed the expression level of a set of imprinted genes and you direct the reader to table 1. However in the table 1 there are not any gene and you do not provided any result of the experiments carried out to fill this section. Could you please provide the data or re-direct to the correct table if there is one?

Author response: Thank you for pointing out the mistake; we modified the text as suggested, citing “table 2” for the imprinted genes and “table 3” for WNT loci.

3) Please, re-write the legend of the table 2 in order to improve its understanding. Although it is written in the main text, it is a bit confusing.

Author response: Thank you for the suggestion, we modified the legend of table 2 to improve its understanding.

4) Why do you not validate the different expression level results of the WNT panel by qPCR? Please do it to positively confirm the results, at least for the most differently expressed genes.        

 Author response: We used Nanostring technology as it represents a medium-throughput platform to evaluate the mRNA abundance profiles, providing a reproducible and fully automated analyses of the samples. The robustness of this technology was already validated as reported in several papers now quoted in the manuscript (see for example references 1-2 reported below).

The reliability of Nanostring technology is based on the ability to quantify the expression of multiple genes without amplification steps. Conversely, in qPCR technical artifacts could be introduced (Ref. 3).

In addition, the aim of these experiments was to assess whether WNT pathways were altered, and to do this we investigated 180 genes starting from the same batch of cells.

Considering all these aspects, we did not perform validation by means of another approach.

To highlight the reliability of Nanostring technology we added the following sentence and the related references in ‘Materials and Methods’ section ‘nCounter Analysis’ paragraph (page 8 lines 191-196) : “We used  Nanostring technology  as it represents a medium-throughput platform to evaluate mRNA abundance profiles providing reproducible and fully automated analyses of the samples. The robustness of this technology was already validated in several papers [1 and 2].  The reliability  of  Nanostring technology is based on the ability to quantify the expression of multiple genes without amplification steps.  Conversely, technical artifacts could be introduced in qPCR [3].”

References that highlight the robustness and the reliability of the Nanostring technology:

  1. Veldman et al., Evaluating Robustness and Sensitivity of the NanoString Technologies nCounter Platform to Enable Multiplexed Gene Expression Analysis of Clinical Samples. Cancer Res. 2015
  2. Gentien et al., Digital Multiplexed Gene Expression Analysis of mRNA and miRNA from Routinely Processed and Stained Cytological Smears: A Proof-of-Principle Study. Acta Cytol. 2021
  3. Prokopec et al., Systematic evaluation of medium-throughput mRNA abundance platforms. RNA. 2013
  4. Theis M, Paszkowski-Rogacz M, Weisswange I, Chakraborty D, Buchholz F. Targeting Human Long Noncoding Transcripts by Endoribonuclease-Prepared siRNAs. J Biomol Screen. 2015 Sep;20(8):1018-26. doi: 10.1177/1087057115583448.
  5. Maxfield KE, Taus PJ, Corcoran K, Wooten J, Macion J, Zhou Y, Borromeo M, Kollipara RK, Yan J, Xie Y, Xie XJ, Whitehurst AW. Comprehensive functional characterization of cancer-testis antigens defines obligate participation in multiple hallmarks of cancer. Nat Commun. 2015 Nov 16;6:8840. doi: 10.1038/ncomms9840. PMID: 26567849; PMCID: PMC4660212.
  6. Chen L, Engel BE, Welsh EA, Yoder SJ, Brantley SG, Chen DT, Beg AA, Cao C, Kaye FJ, Haura EB, Schabath MB, Cress WD. A Sensitive NanoString-Based Assay to Score STK11 (LKB1) Pathway Disruption in Lung Adenocarcinoma. J Thorac Oncol. 2016 Jun;11(6):838-49. doi: 10.1016/j.jtho.2016.02.00
  7. Richard AC, Lyons PA, Peters JE, et al. Comparison of gene expression microarray data with count-based RNA measurements informs microarray interpretation. BMC Genomics. 2014;15(1):649. Published 2014 Aug 4. doi:10.1186/1471-2164-15-649
  8. Rooney C, Geh C, Williams V, et al. Characterization of FGFR1 Locus in sqNSCLC Reveals a Broad and Heterogeneous Amplicon. PLoS One. 2016;11(2):e0149628. Published 2016 Feb 23. doi:10.1371/journal.pone.0149628

5) Have you analyzed the data from the WNT panel splitting by gene? NIPBL mutated vs SMC1A mutated samples? In the already published manuscript of Boudaoud I et al from 2017 (Genetics, 2017 Sep;207(1):139-151. doi: 10.1534/genetics.117.202291) they reported that there are only 126 differently expressed genes shared between NIPBL and SMC1A in mutated LCLs derived samples. Please re-analyze our data in this way in order to confirm that these genes are truly dysregulated in both types of samples.         

Author response: Thank you, your suggestion to re-analyse our data, and a careful reading of the paper by Boudaoud et al. allowed us to improve the quality of our findings. We performed additional expression analyses considering three subsets of samples: NIPBL-mutated CdLSs vs. CTRL, SMC1A-mutated CdLSs vs. CTRLs and NIPBL- vs. SMC1A- mutated CdLSs. We reported these data in figure S2 (supplementary material).

In particular, we added the following considerations to our manuscript:

-Results (page 23 lines 393-410) ‘Finally, we performed three different expression analyses splitting CdLSs samples based on NIPBL and SMC1A involvement, comparing NIPBL-mutated CdLSs vs. CTRLs (figure S2a), SMC1A-mutated CdLSs vs. CTRLs (figure S2b) and NIPBL- vs. SMC1A- mutated CdLSs (figure S2c).  Differently to the main cohort (CdLSs vs CTRL, figure 5 and 6), in none of the three subsets, we found differentially expressed genes with Benjamini-Hochberg adjusted p-values <0.1. Some genes reached a significant unadjusted p-value < 0.005. Interestingly, considering the NIPBL-mutated samples, 17 DEGs reached the significant threshold of 0.005, among them nine are shared with the most significant DEGs of the main cohort: SFRP1, TCF4, GNAS, MEST, MAPK10, CDKN2A, PLCB1, RUNX2 and MMP7 (figure S2a). Similarly, the SMC1A-mutated group showed nine out of the 25 significant DEGs in common with the main cohort: RUNX2, BMP4, MAPK10, GNAS, CDKN2A, TGFB3, TCF4, PEG10 and PLGB1 (figure S2b). When we consider the NIPBL- vs. SMC1A- mutated CdLSs subset, the number of shared significant DEGs (p-value < 0.005) drops to 2: RUNX2 and MAPK10 (figure S2c). These findings are in line with the results of Boudaoud et al. that compared the gene expression profiles of LCLs from patients carrying mutations in NIPBL and SMC1A and showed reduced number of differentially expressed genes shared between the two gene groups (n= 126) compared to the totality of genes misregulated in NIPBL-mutated (n= 1431) and SMC1A-mutated (n=1186) samples  [Boudaoud I et al from 2017].

- Discussion (page 27 lines 493-495): “Interestingly, RUNX2 and MAPK10 are the only two significant DEGs (unadjusted p-value 0.0462 and 0.0221, respectively) when we consider the NIPBL- vs SMC1A- mutated CdLSs subset”.

6) I am aware of the difficulty to obtain biological samples of a different origin in CdLS patients; however it will improve the quality of your data if you could confirm the results of some of these experiments in those different samples.      

Author response: We agree that the evaluation of different tissues from patients should be very useful. Unfortunately, we obtained these cell lines from the biobank, “The Galliera Genetic Bank”, member of the Telethon Network of Genetic Biobanks in which no other source of biological material was available. To extend the study to different tissues we have to recruit CdLS patients and obtain a new Ethics committee approval. At the moment we haven’t other patients/tissues.      

Reviewer 2 Report

The effect of cohesin in stabilizing this locus in human cells was previously established (Nativio and coworkers, reference 29).  In this case, the authors investigate the effect of genetic variations of cohesin genes on this locus in lymphoblastoid cell lines derived from CdLS patients. To do that, they use a different approach with four anchors and the 3C results they get are very clearly summarized in interactome schemes.  Unfortunately, the genes in this locus are not expressed in the cell lines used in the manuscript, so the perturbations in chromatin configuration cannot be directly associated with changes in gene expression. However, they also expanded their research to the expression of other genes , including genes known to be often regulated by high order chromatin arrangements and genes in the WNT pathway, which seem to be transcriptionally altered. They also associate methylation instability in these loci with cohesin defects, opening interesting possibilities to further research.

The work seems solid, well conducted and designed, and rigorous. It do not only confirms in a different model but also expands what was previously known from previous publications.

I would like to share some major and minor comments that I think might reinforce the work and make it suitable for publication.

Comments:

-In the abstract (line 22) it is stated that  CdLS is “caused by constitutive mutations in cohesin complex genes”. Then, in line 63, only NIPBL, SMC1A, SMC3, RAD21 and HDAC8 are considered as “causative genes” for CdLS. However, even if it’s clear that most of the analyzed cases of CdLS are associated with mutations in NIPBL, several other genes coding for factors not directly linked to cohesin function have been recently linked to CdLS. This has been recently reviewed in several works. In light of this accumulated evidence on non-cohesin factors involvement in CdLS ethiology, I think that the affirmations in the text might be revised.

-In the first paragraph of results (lines 216 and 217), a 3C analysis in four CdLS LCLs (lines CdLS1, 2, 6 and 9) is described, but only the results for three of them are shown (CdLS1 in Figure 2; CdLS2 in Figure 3; and CdLS6 in Figure 4). I may have missed it, but I did not find any explanation for this omission in the text. I wonder if this could be justified or, alternatively, if the results obtained for the CdLS9 LCL could be also shown and discussed. This is particularly interesting, since the mutation in this omitted cell line affects NIPBL, which is known as the main causative factor by far for CdLS (lines 64 and 65).

-Authors state that cohesin mutations cause chromatin conformation perturbations in the H19/IGF2 locus. However, in order to strengthen the hypothesis on a direct effect on the structure of this locus when cohesin subunits are mutated, it might be interesting to analyze the binding of cohesin to relevant elements of this locus. A ChIP experiment on a core cohesin subunit (as RAD21, for  example) on some CTCF binding sites of the H19/IGF2 locus, might shed some light on the association of cohesin with this structure. The comparison with the ChIP signal in control LCLs will tell us if mutations associated with CdLS alter the association of cohesin to these loci. If such an alteration is confirmed, this could be hypothesized as the cause for the altered conformation of the locus, and so a direct effect of cohesin on the perturbation of this imprinted region could be considered.

-Minor comments:

-Some erratum should be corrected, as “lymphoblastoid” in line 110, “cohesin” in line 267, or “Nanostring” in line 269.

-In the discussion, the affirmation that “our results confirm the involvement of the WNT pathways in CdLS development”(line 408) seems to go a bit far. To me, the results presented in the manuscript show that genes associated with the WNT pathway display altered expression in LCLs derived from CdLS patients. I don´t think that this is exactly a confirmation of the involvement of the WNT pathways in CdLS development, since this transcriptional perturbance could be more a consequence of CdLS development. I gently suggest the authors to re-think on this statement, to see if they are sure that is what they mean.

-It seems to me that the effect observed in reference 29 on the H19/IGF2 locus upon cohesin knock-down (reduction in looping interaction) could be seen as opposite to the effect detected in this manuscript in cohesin mutant cell lines (the predominant appearance of new contacts). If that is the case, I think it should be discussed in the discussion section.

Author Response

1) In the abstract (line 22) it is stated that  CdLS is “caused by constitutive mutations in cohesin complex genes”. Then, in line 63, only NIPBL, SMC1A, SMC3, RAD21 and HDAC8 are considered as “causative genes” for CdLS. However, even if it’s clear that most of the analyzed cases of CdLS are associated with mutations in NIPBL, several other genes coding for factors not directly linked to cohesin function have been recently linked to CdLS. This has been recently reviewed in several works. In light of this accumulated evidence on non-cohesin factors involvement in CdLS ethiology, I think that the affirmations in the text might be revised.

Author response: Thank you for your useful observation; we modified the Introduction paragraph following your suggestions. In particular, we added the sentence: ‘In addition to the abovementioned cohesin core complex alterations, mutations in other genes, such as AFF4, ANKRD11, CREBBP, and EP300, have been identified in patients with a phenotype resembling CdLS [9]. (page 3 lines 75-77)

2) In the first paragraph of results (lines 216 and 217), a 3C analysis in four CdLS LCLs (lines CdLS1, 2, 6 and 9) is described, but only the results for three of them are shown (CdLS1 in Figure 2; CdLS2 in Figure 3; and CdLS6 in Figure 4). I may have missed it, but I did not find any explanation for this omission in the text. I wonder if this could be justified or, alternatively, if the results obtained for the CdLS9 LCL could be also shown and discussed. This is particularly interesting, since the mutation in this omitted cell line affects NIPBL, which is known as the main causative factor by far for CdLS (lines 64 and 65).

Author response: Thanks for your comment. We did the 3C analysis on all the four cell lines indicated. CdLS9 results are commented in the text and shown in supplementary figure S1. The other cell line carrying NPBL variant was the CdLS6, whose results are described in the text and shown in figure 4.

3) Authors state that cohesin mutations cause chromatin conformation perturbations in the H19/IGF2 locus. However, in order to strengthen the hypothesis on a direct effect on the structure of this locus when cohesin subunits are mutated, it might be interesting to analyze the binding of cohesin to relevant elements of this locus. A ChIP experiment on a core cohesin subunit (as RAD21, for  example) on some CTCF binding sites of the H19/IGF2 locus, might shed some light on the association of cohesin with this structure. The comparison with the ChIP signal in control LCLs will tell us if mutations associated with CdLS alter the association of cohesin to these loci. If such an alteration is confirmed, this could be hypothesized as the cause for the altered conformation of the locus, and so a direct effect of cohesin on the perturbation of this imprinted region could be considered.

Author response: We agree that the ChIP approach can be used to understand how mutated cohesin alter the association with their target loci. Unfortunately, based on the complexity of ChIP approach we cannot perform it for this study. We will consider the referee's suggestion for a next study on this topic.

-Minor comments:

4) Some erratum should be corrected, as “lymphoblastoid” in line 110, “cohesin” in line 267, or “Nanostring” in line 269.

Author response:  Thank you for pointing out these mistakes. The terms were corrected in the text

5) In the discussion, the affirmation that “our results confirm the involvement of the WNT pathways in CdLS development”(line 408) seems to go a bit far. To me, the results presented in the manuscript show that genes associated with the WNT pathway display altered expression in LCLs derived from CdLS patients. I don´t think that this is exactly a confirmation of the involvement of the WNT pathways in CdLS development, since this transcriptional perturbance could be more a consequence of CdLS development. I gently suggest the authors to re-think on this statement, to see if they are sure that is what they mean.

Author response: We truly appreciate your comment and modified the paragraph following your suggestion, as follows: “Our results confirm alterations of the WNT pathways in LCLs derived from CdLS patients; in particular, we observed the most significant differences in the canonical WNT pathway, cell cycle and WNT signal negative regulation.” (page 22 lines 375-377 and page 27 lines 503-505)

6) It seems to me that the effect observed in reference 29 on the H19/IGF2 locus upon cohesin knock-down (reduction in looping interaction) could be seen as opposite to the effect detected in this manuscript in cohesin mutant cell lines (the predominant appearance of new contacts). If that is the case, I think it should be discussed in the discussion section.

Author response: Thanks for the tip. We have carefully analysed the paper by Nativio et al. and we think that the differences between the two studies are probably related to the knockdown of the cohesin proteins obtained by RNAi in Nativio et al. experiments, compared to our approach       performed on cells carrying heterozygous mutations, implying a residual/altered function of the cohesin. In addition, based on the information in Nativio paper (fig 1) the coverage of our 3C experiment seems to be higher. 

Besides, it can be hypothesized that a complete ablation of cohesin subunit SCC1/RAD21 obtained by RNAi determines the general reduction of the interaction strength. Differently, in our LCLs, in which the mutation affects one allele, a malfunction  is expected, rather than the lack of the cohesin function. We clarify this concept in the discussion (page 25-26 lines 446-454): “We found a broad perturbation in the chromatin structure of the domain regardless of the CdLS causative gene, and observed a change in the interactions among CTCF-binding sites, regulatory elements and genes of the region. This scenario could be related to the existence in each cell line of a diffuse instability rather than recurrent alterations of specific interactions. It is also conceivable that compromised maintenance of chromatin architecture could lead to heterogeneous defects among cells in the same cell line. The alterations can be due to cohesin malfunction rather than lack of function. The improper sliding of the cohesin complex along the loop could, indeed, causes the observed perturbed chromatin interactions, including new associations.”

Round 2

Reviewer 2 Report

Although authors have not performed the suggested ChIP experiments, which would have been informative, they have followed the rest of suggestions on text modifications and the manuscript has improved.